# HFSTI-Net: Hierarchical Frequency-spatial-temporal Interactions for Video Polyp Segmentation

**Yuanqin He[1], Guilian Chen[1], Yuhua Zhang[1], Huisi Wu[1]\*, Jing Qin[2]**
[1]College of Computer Science and Software Engineering, Shenzhen University
[2]Centre for Smart Health, School of Nursing, The Hong Kong Polytechnic University
`2400101064@mails.szu.edu.cn, hswu@szu.edu.cn`

## ABSTRACT

Automatic video polyp segmentation (VPS) is crucial for preventing and treating colorectal cancer by ensuring accurate identification of polyps in colonoscopy examinations. However, its clinical application is hampered by two key challenges: shape collapse, which compromises structural integrity, and episodic amnesia, which causes instability in challenging video sequences. To address these challenges, we present a novel video segmentation network, *HFSTI-Net*, which integrates global perception with spatiotemporal consistency in spatial, temporal, and frequency domains. Specifically, to address shape collapse under low contrast or visual ambiguity, we design a Hierarchical Frequency-spatial Interaction (HFSI) module that fuses spatial and frequency cues for fine-grained boundary localization. Furthermore, we propose a recurrent mask-guided propagation (RMP) module that introduces a dual enhancement mechanism based on feature memory and mask alignment, effectively incorporating spatiotemporal information to alleviate inter-frame inconsistencies and ensuring long-term segmentation stability. Extensive experiments on the SUN-SEG and CVC-612 datasets demonstrate that our method achieves real-time inference and outperforms other state-of-the-art approaches. Codes are available at `https://github.com/Yuanqin-He/HFSTI-Net`.

## 1 INTRODUCTION

Colorectal cancer (CRC), a prevalent gastrointestinal malignancy and the third most common cancer globally, can be effectively prevented Shaukat & Levin (2022) through timely screening and removal of precursor polyps via colonoscopy Wu et al. (2024). However, the diagnostic process heavily relies on endoscopists' expertise, and hence inexperienced practitioners risk missing the detection of precancerous lesions. Therefore, accurate and real-time automated polyp segmentation is crucial for enhancing early CRC diagnosis and supporting timely clinical decision-making.

In recent years, numerous deep learning based methods have been proposed for image polyp segmentation (IPS) and achieved remarkable successes Wei et al. (2021); Dong et al. (2021); Zhou et al. (2023). However, these approaches still face two inherent limitations in real-world clinical settings. First, as illustrated in Figure 1 (b), the low contrast between polyps and the surrounding mucosa makes it challenging to accurately distinguish the target from the background only based on static image information alone Fan et al. (2020b); Wu et al. (2023). This often leads to a phenomenon that we usually call *shape collapse* in the segmentation results. Second, these static image-based methods overlook a critical fact that real-world clinical screening is conducted based on a continuous video stream Puyal et al. (2020); Ji et al. (2021); Li et al. (2022). In the video, the appearance of polyps can undergo drastic changes due to variations in viewpoint, intestinal peristalsis, and camera motion (as shown in Figure 1 (c)), where appearance not only refers to visual texture but also encompasses substantial variations in polyp size, position, and shape. While these temporal variations pose a new challenge for accurate segmentation, they are also probably helpful to address the

---
\*Corresponding Author

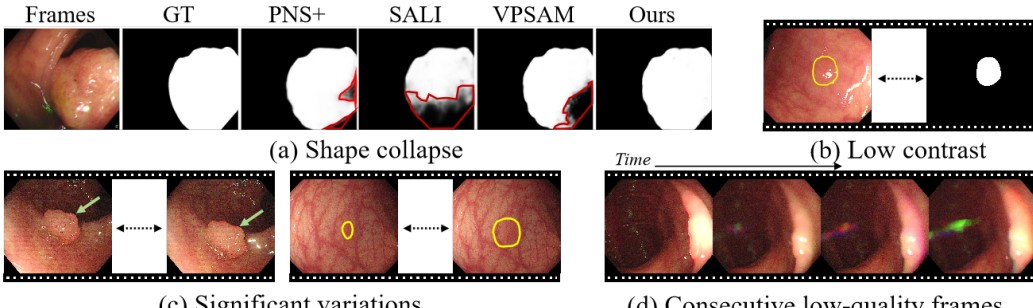

Figure 1: Challenges of VPS including (a) shape collapse, (b) low contrast between polyps and background, (c) significant variations between adjacent frames, and (d) consecutive low-quality frames.

first limitation, as different frames can provide complementary views to disambiguate low-contrast regions. Existing IPS methods, however, are inherently not able to sufficiently leverage temporal information, and hence cannot achieve satisfactory segmentation results in clinical practice. To more closely align with real-world clinical scenarios, the task of video polyp segmentation (VPS) has emerged  Ji et al. (2022); Hu et al. (2024). VPS methods aim to enhance segmentation robustness and accuracy by leveraging temporal coherence in videos. Current mainstream VPS approaches, either employing hybrid 2D/3D architectures  Puyal et al. (2020); Bhattacharya et al. (2024); Xu et al. (2024) or harnessing self-attention mechanisms  Vaswani (2017); Ji et al. (2021); Chen et al. (2024), predominantly rely on implicit modeling of dense, pixel-level features to propagate temporal information. However, these methods are highly sensitive to appearance variations as pixel-level features largely lack high-level semantic abstraction. Consequently, these methods cannot address challenges caused by large temporal gaps among frames, such as drastic deformations and sequences of low-contrast and blurry frames (as shown in  Figure 1 (d)), which lead to the *episodic amnesia*. To address this shortcoming, some approaches propose to employ global-to-local learning to capture more temporal information, but these methods are usually unstable as the temporal information they extracted is quite limited.

Although VPS improves robustness with temporal coherence, it still fails under low contrast or occlusion, leading to structural loss and missed polyps. Recently, integrating frequency information into deep learning models has shown considerable promise for many computer vision tasks, such as camouflaged object detection (COD) Fan et al. (2020a). Existing spatial-domain methods, such as those focusing on enhancing pixel-level discriminability Ji et al. (2021); Cheng et al. (2022); Wang et al. (2022), often struggle with capturing comprehensive contextual information due to their inherent limitations in local feature representation. Accordingly, for dealing with 'camouflaged' polyps in colon structures, leveraging global frequency perception and harnessing multi-domain representations may enable the model to perceive spatial details within features. While conventional spatial-domain methods Chi et al. (2020); Wang et al. (2023b) struggle to effectively identify targets from the background due to their high visual similarity, recent works have begun exploring frequency-domain representations to enhance identification capacity. To address this issue, recent works Li et al. (2024); Wang et al. (2023a) explore the frequency-domain representations to enhance the identification capacity. However, these methods ignore the modeling of frequency-spatial interactions, limiting their ability to adaptively fuse cross-domain features and ultimately hindering spatial-domain feature learning. Besides, these methods fail to model the dependencies between different frequency components and their spatial counterparts, leading to incomprehensive feature representations for precise target localization.

In this paper, we propose a novel network, *HFSTI-Net*, for VPS, addressing challenges by jointly modeling frequency, spatial, and temporal features. The network consists of two key components: a hierarchical frequency-spatial interaction (HFSI) module and a recurrent mask-guided propagation (RMP) module. The HFSI module uses a dual-path design to combine local spatial cues with global frequency representations, effectively capturing fine-grained boundary details and preventing shape collapse, even in low-contrast or ambiguous conditions. The RMP module introduces a memory-based dual enhancement mechanism, storing historical embeddings and predictions to model temporal dependencies and mitigate inconsistencies, improving spatiotemporal consistency

and reducing episodic amnesia. Extensive experiments on SUN-SEG and CVC-612 demonstrate that *HFSTI-Net* outperforms SOTA methods. Our contributions are summarized as follows:

- We propose a novel video polyp segmentation network, *HFSTI-Net*, that jointly models spatial, frequency, and temporal information to address challenges such as background interference, low contrast, and rapid endoscope movements.

- We design a HFSI module to enhance boundary localization and structural integrity by integrating local spatial cues with global frequency representations through a dual-path Fourier-based interaction. Furthermore, we propose a RMP module that leverages feature memory and mask alignment to capture long-term spatiotemporal dependencies, effectively alleviating episodic amnesia and reducing tracking errors.

- We conduct extensive experiments on the SUN-SEG and CVC-612 dataset, which demonstrates that our method achieves superior performance compared to other SOTA methods but also maintains real-time efficiency, making it more suitable for clinical deployment.

## 2 RELATED WORKS

### 2.1 POLYP SEGMENTATION

With the development of deep learning, remarkable progress has been made in IPS. Although IPS primarily leverages CNN architectures Cheng et al. (2021); Wu et al. (2022) for local feature extraction, their limited global perspective leads to blurred boundaries and low-contrast issues. To address this, Transformers Vaswani (2017) or hybrid CNN-Transformer architectures Zhang et al. (2021); Li et al. (2021) were introduced to enhance global context awareness, alongside boundary constraint methods Fan et al. (2020a); Cheng et al. (2021) for edge refinement. However, VPS requires modeling temporal dynamics, an inherent limitation of static IPS methods. Early methods used hybrid 2D/3D convolutions to fuse spatial and short-term temporal features Puyal et al. (2020), but their limited receptive field struggles with long-range dependencies. To address this, attention-based methods emerged Ji et al. (2021), with models like PNS+ Ji et al. (2022) capturing global temporal relationships across entire sequences. While effective, global attention is often computationally expensive and sensitive to noise. To improve efficiency, later works introduced key frame-guided strategies Xu et al. (2022); Hu et al. (2024), using high-quality frames as anchors. This highlights that most temporal models implicitly learn inter-frame relationships via end-to-end training. To address this limitation and mitigate episodic amnesia in challenging video sequences, we propose the RMP module. Leveraging high-level features and predicted masks from a memory bank, RMP captures rich spatiotemporal cues, allowing the model to recall dynamic changes and historical context across frames. This enhances polyp recognition and localization while ensuring long-term segmentation stability.

### 2.2 FREQUENCY LEARNING

The frequency domain plays an important role in signal analysis and has been increasingly applied to computer vision tasks. Several methods He et al. (2023); Wang et al. (2023a) leverage frequency information for enhanced feature representation. FcaNet Qin et al. (2021) views channel attention as a frequency-domain compression problem and introduces multi-spectral attention to preserve informative frequency features with no added complexity. Wang et al. (2023b) investigate frequency learning in segmentation, showing that networks tend to focus on class-specific frequency components, which may lead to frequency shortcuts and hinder generalization. To address the camouflage challenge, FAGF-Net Li et al. (2024) incorporates frequency-aware attention and graph-based fusion, outperforming spatial-only methods. However, existing frequency-based methods often treat frequency and spatial information somewhat independently or lack a deep integration. This can lead to an incomplete capture of intricate relationships between different frequency components and their precise spatial counterparts, potentially limiting its effectiveness in the task of VPS. Therefore, we propose a novel HFSI module designed to counteract shape collapse so enhance the structural integrity of segmented polyps. It allows for exploring fine-grained structural details by learning and enhancing frequency-spatial interactions from the frequency domain to distinguish foreground polyp targets from surrounding tissues.

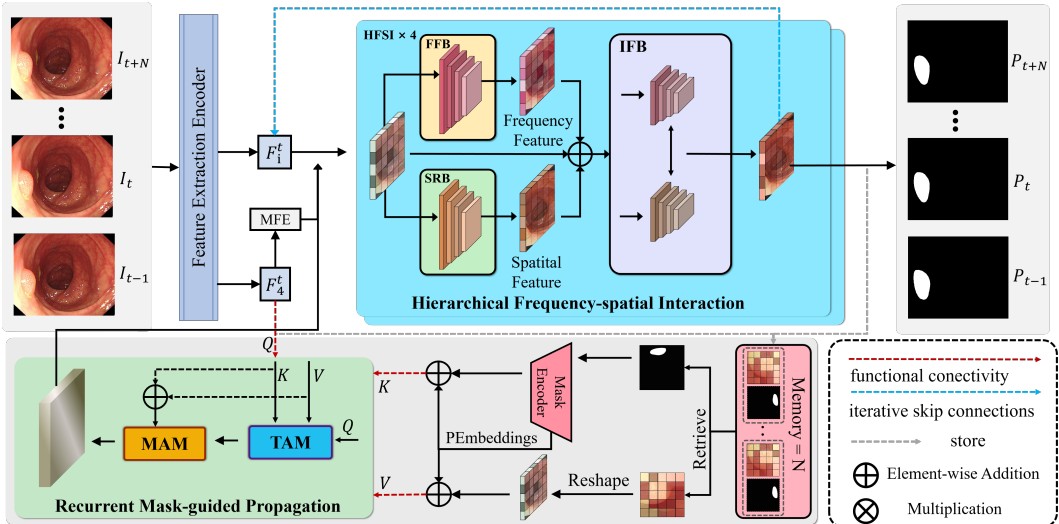

Figure 2: Overview of the proposed network for VPS. Our model addresses shape collapse and episodic amnesia to ensure structural integrity and long-term segmentation stability. The HFSI module processes features in both spatial and frequency domains for fine-grained boundary localization, counteracting shape collapse. The RMP module utilizes a memory bank and cross-attention for consistent tracking, mitigating episodic amnesia. MFE blocks, with parallel convolutions (1×1, 3×3, 5×5), extract rich spatial features for complementary enhancement.

## 3 METHOD

### 3.1 OVERVIEW

Figure 2 presents the overall framework of our method, which addresses two key challenges in VPS: maintaining structural integrity in segmented polyps and ensuring long-term segmentation stability across challenging video sequences. To address shape collapse and preserve structural integrity, we introduce a Hierarchical Frequency-spatial Interaction (HFSI) module. By jointly processing features in the spatial and frequency domains and fusing them through an interwoven dual-path design, it combines local detail cues with global context, enabling fine-grained boundary localization and robust shape preservation, even under visual ambiguity and low contrast. To enforce long-term segmentation stability and mitigate episodic amnesia in challenging video sequences, we propose a Recurrent Mask-guided Propagation (RMP) module. It stores previous frame features and masks in a memory bank and retrieves relevant information via cross-attention. A mask affinity mechanism further aligns the current prediction with historical context, ensuring consistent and stable tracking of polyps across frames, even through occlusions or significant appearance changes.

As shown in Figure 2, given an input video sequence $\{I_t\}_{t=1}^T$ with $I_t \in \mathbb{R}^{H \times W \times 3}$, we first extract multi-level features $F = \{F_i^t\}_{i=1}^4$ using a backbone encoder, where each $F_i^t$ has spatial resolution $\frac{H}{2^{i+1}} \times \frac{W}{2^{i+1}}$. Then, the top-level feature $F_4^t$ is enhanced by a MFE module. To model temporal coherence, the current feature $F_4^t$ and historical context $F_4^{t-1}$, $P_{t-1}$ are passed into the RMP module. The refined temporal feature is further processed by the HFSI module, which integrates local spatial and global frequency cues through interwoven fusion, yielding enriched representations $X = \{\chi_i\}_{i=1}^4$. Finally, a decoder aggregates these features and progressively refines them to generate the prediction $P = \{P_t^i\}_{i=1}^4$.

### 3.2 HIERARCHICAL FREQUENCY-SPATIAL INTERACTION

Video polyp segmentation (VPS) is challenging due to the high visual similarity between polyps and surrounding tissues in color, texture, and motion blur. While spatial-domain methods focus on local details, they often miss global context. In contrast, frequency-based approaches capture global semantics but are susceptible to noise and may underutilize informative frequency cues in complex scenes. To address these issues, we propose the HFSI module with a dual-path structure

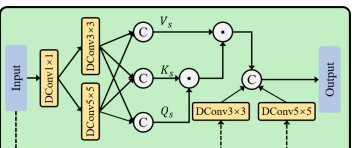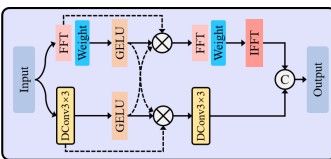

(a) Frequency Filter Block (FFB)  (b) Spatial Refinement Block (SRB)  (c) Interwoven Fusion Block (IFB)

Figure 3: The key details module of the hierarchical frequency-spatial interaction (HFSI) module. The module mainly consists of frequency filter block (FFB), spatial refinement block (SRB) and interwoven fusion block (IFB).

that extracts both spatial and frequency representations. Central to HFSI is an *interwoven fusion mechanism*, which facilitates fine-grained interaction between the two domains, enabling adaptive enhancement of meaningful features. As shown in Figure 3, the HFSI module consists of three components: the Frequency Filter Block (FFB), which extracts global contextual patterns through frequency-domain self-attention to enhance boundary localization and suppress background noise; the Spatial Refinement Block (SRB), which emphasizes edge-aware details to preserve local structural precision; and the Interwoven Fusion Block (IFB), which adaptively blends frequency- and spatial-domain features via a learnable attention mechanism for semantic alignment. By integrating spatial and frequency cues across multiple levels, HFSI effectively improves both local detail and global context, enabling more accurate segmentation of camouflaged polyps in complex scenes.

### 3.2.1 FREQUENCY FILTER BLOCK

To extract global contextual patterns in the frequency domain, the FFB applies a frequency-domain self-attention mechanism that models channel-wise dependencies across spectral components. By emphasizing salient frequency responses and suppressing redundancy, it sharpens polyp boundaries and reduces background interference, especially in low-contrast frames. Given an input feature $\mathcal{X} \in \mathbb{R}^{C \times H \times W}$, composed of current-layer and high-level features, we first apply layer normalization to obtain $\hat{\mathcal{X}} = \text{LN}(\mathcal{X})$. The normalized feature is then transformed into the frequency domain via Fast Fourier Transform (FFT) to compute the query, key, and value: $Q_f, K_f, V_f = \mathcal{F}^{q,k,v}(\hat{\mathcal{X}})$, respectively. We compute the frequency attention map via matrix multiplication and apply it to reweight $V_f$, followed by inverse FFT to restore the spatial representation. In parallel, we introduce a frequency residual branch that enriches spectral responses through a lightweight attention filter $\sigma(\cdot)$ containing convolution, normalization, and activation layers. Finally, the outputs from both branches are concatenated to yield the enhanced frequency-aware feature:

$$X_f^r = \text{Cat}\left(\mathcal{F}^{-1}(\Lambda_f \odot V_f), \ \mathcal{F}^{-1}(\sigma(\mathcal{F}(\hat{\mathcal{X}})))\right), \tag{1}$$

where $\Lambda_f = Q_f \odot K_f$ denotes the attention map and $\odot$ represents matrix multiplication.

This design ensures that rich global frequency cues are preserved and fused with spatial cues in later modules, improving the network's robustness to boundary ambiguity and structural complexity in polyp regions.

### 3.2.2 SPATIAL REFINEMENT BLOCK

To accurately segment polyps with varying sizes and complex shapes, we design the Spatial Refinement Block (SRB) to capture fine-grained structural details and local context. Unlike frequency-based modeling, SRB operates entirely in the spatial domain and emphasizes edge-sensitive features through spatial self-attention. As illustrated in Figure 3 (b), the input feature $\mathcal{X}$ is first passed through a $1 \times 1$ convolution to encode positional information. To effectively capture multi-scale spatial dependencies, we employ two depthwise separable convolutions with kernel sizes $3 \times 3$ and $5 \times 5$, respectively. These are used to compute the query, key, and value representations:

$$Q_s, \ K_s, \ V_s = \text{Cat}(\mathcal{DC}_3^{q,k,v}(\mathcal{X}), \ \mathcal{DC}_5^{q,k,v}(\mathcal{X})), \tag{2}$$

The spatial attention map is computed as $\Lambda_s = \text{Softmax}(Q_s \odot K_s)$, which highlights salient spatial structures. We then use this attention to reweight $V_s$, yielding a refined attention output. In parallel, a residual spatial branch $\mathcal{X}_s^r = \text{Cat}(\mathcal{DC}_3(\mathcal{X}), \mathcal{DC}_5(\mathcal{X}))$ is introduced to preserve original spatial

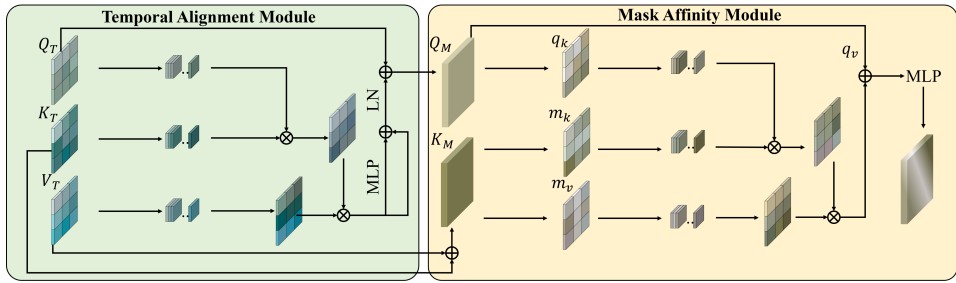

Figure 4: Structure of the recurrent mask-guided propagation module (RMP). The module consists of temporal alignment module (TAM) and mask affinity module (MAM).

details. The final spatial-aware feature is formed by concatenating both branches:

$$X_s^r = \text{Cat}(\Lambda_s \odot V_s, \ \mathcal{X}_s^r). \tag{3}$$

### 3.2.3 INTERWOVEN FUSION BLOCK

Frequency and spatial features encode complementary information: frequency features emphasize global semantic distributions, while spatial features highlight fine-grained structural details. To fully leverage the strengths of both representations, we design the Interwoven Fusion Block (IFB) as a learnable bridge that enables dynamic feature entanglement across the two domains.

Given frequency-domain features $X_f$, spatial-domain features $X_s$, and an earlier-layer feature $X$, we first compute a composite representation via residual fusion: $X_c = X_f + X_s + X$. We then apply layer normalization to obtain the normalized feature $\hat{X}_c = \text{LN}(X_c)$, which serves as input to a two-stage inter-domain interaction process. We separately enhance frequency and spatial representations by projecting $\hat{X}_c$ into each domain and applying gated multiplicative attention:

$$\hat{X}_f^2 = \text{GeLU}(\sigma(\mathcal{F}(\hat{X}_c)) \otimes \sigma(\mathcal{F}(\hat{X}_c)), \tag{4}$$

$$\hat{X}_s^2 = \text{GeLU}(\mathcal{DC}_3(\hat{X}_c)) \otimes \mathcal{DC}_3(\hat{X}_c), \tag{5}$$

where $\mathcal{F}(\cdot)$ and $\mathcal{DC}_3(\cdot)$ denote the fast Fourier transform and depthwise convolution, respectively, and $\sigma(\cdot)$ is a lightweight channel-wise attention filter. We begin by combining the intermediate frequency and spatial features via element-wise multiplication: $\hat{X}_{fs} = (\hat{X}_f^2 \otimes \hat{X}_s^2)$, which serves as the input to both the frequency and spatial refinement branches. The frequency branch performs gated attention in the Fourier domain, while the spatial branch focuses on localized filtering. The outputs of both branches are then aggregated:

$$\hat{X}_f^3 = \mathcal{F}^{-1}\left(\sigma(\mathcal{F}(\hat{X}_{fs})) \otimes \mathcal{F}(\hat{X}_{fs})\right), \tag{6}$$

$$\hat{X}_s^3 = \mathcal{DC}_3(\hat{X}_{fs}), \quad \hat{X}_c^3 = \text{Cat}(\hat{X}_f^3, \hat{X}_s^3) + X_c. \tag{7}$$

Through interwoven, cross-domain interaction with gated fusion, IFB aligns global and local features, reducing misalignment and background ambiguity for more precise segmentation.

### 3.3 RECURRENT MASK-GUIDED PROPAGATION MODULE

Although HFSI improves structural integrity, VPS still suffers from temporal inconsistency caused by blur, motion, occlusion, and appearance changes. To address this, we propose the RMP module, which explicitly models spatiotemporal dependencies to ensure long-term consistency. RMP maintains a memory bank of high-level features and masks from past frames. For each incoming frame, current features act as a query, while memory features and masks serve as key-value pairs. Temporal alignment module integrates temporal cues, and the result is refined with MLP and normalization. To further enhance alignment, a mask affinity module fuses the predicted mask with spatial features and performs another cross-attention step with memory, enabling motion-consistent polyp localization over time.

To model spatiotemporal dependencies, we employ a cross-attention mechanism within the temporal alignment module. Given the current frame feature $Q_T$ as the query and the memory bank features

Table 1: Quantitative comparison with different state-of-the-art methods on SUN-SEG and CVC-612 test sets. The highest value is indicated in bold, while the second highest value is underlined.

| Model | Backbone | Class | SUN-SEG-Easy | | | | SUN-SEG-Hard | | | | CVC-612 | | | |
|---|---|---|---|---|---|---|---|---|---|---|---|---|---|---|
| | | | $S_\alpha$ | $E_\phi^{mn}$ | $F_\beta^w$ | Dice | $S_\alpha$ | $E_\phi^{mn}$ | $F_\beta^w$ | Dice | $S_\alpha$ | $E_\phi^{mn}$ | $F_\beta^w$ | Dice |
| ZoomNext | PVT-B2 | NVS | 88.33 | 90.48 | 80.66 | 85.49 | 87.64 | 90.84 | 80.25 | 83.51 | 94.66 | 97.83 | 92.45 | 93.17 |
| SLTnet | PVT-B2 | NVS | 88.13 | 91.75 | 83.09 | 85.91 | 87.04 | 90.89 | 80.98 | 83.36 | 94.84 | 97.37 | 92.73 | 93.62 |
| AutoSAM | VIT-B | IPS | 86.28 | 91.67 | 78.36 | 81.28 | 83.57 | 89.93 | 73.59 | 77.37 | 91.52 | 95.38 | 87.47 | 88.73 |
| WeakPolyp | PVT-B2 | IPS | 89.04 | 92.77 | 83.83 | 85.27 | 88.41 | 92.57 | 82.93 | 84.59 | 91.44 | 95.78 | 88.54 | 88.79 |
| PNS+ | Res-50 | VPS | 86.20 | 86.17 | 76.28 | 82.23 | 84.29 | 86.13 | 72.98 | 79.60 | 94.81 | 96.75 | 89.63 | 93.06 |
| MAST | PVT-B2 | VPS | 84.53 | 89.81 | 77.04 | 78.43 | 86.17 | 91.42 | 77.76 | 80.32 | 92.03 | 95.38 | 87.47 | 90.84 |
| VPSAM | VIT-B | VPS | 89.31 | 92.34 | 82.86 | 85.62 | 88.93 | 92.13 | 82.98 | 85.28 | 93.26 | 95.75 | 89.63 | 92.33 |
| SALI | PVT-B2 | VPS | 89.54 | 93.07 | 83.68 | 86.17 | 87.58 | 91.93 | 80.56 | 83.87 | 91.73 | 95.21 | 86.54 | 88.77 |
| **Ours** | PVT-B2 | VPS | **90.73** | **94.86** | **85.82** | **88.03** | **89.63** | **93.92** | **83.26** | **86.27** | **95.02** | **98.46** | **93.58** | **94.31** |

Table 2: Efficiency comparison with SOTA methods on SUN-SEG.

| Method | SUN-SEG-Hard | | | |
|---|---|---|---|---|
| | Dice | GFlops | Param.(M) | FPS |
| SLTNet | 83.36 | 32.47 | 25.79 | 12.36 |
| ZoomNext | 83.51 | 42.95 | 28.18 | 9.69 |
| PNS+ | 79.60 | 45.99 | 9.79 | 76.08 |
| SALI | 83.87 | 21.19 | 26.14 | 18.07 |
| Ours | 86.27 | 46.77 | 28.53 | 31.27 |

Table 3: The performance of different frame rates on the RMP module.

| label | SUN-SEG-Hard | | | | |
|---|---|---|---|---|---|
| | Dice | IoU | GFlops | Param.(M) | FPS |
| 1-frame | 86.27 | 75.86 | 46.77 | 28.53 | 31.27 |
| 2-frame | 86.41 | 76.07 | 46.91 | 28.85 | 29.43 |
| 3-frame | 86.55 | 76.29 | 47.06 | 29.16 | 28.77 |
| 4-frame | 86.66 | 76.46 | 47.19 | 29.48 | 26.94 |

$K_T$ and $V_T$ as key and value, we compute the attention-enhanced representation as:

$$Z = L(\text{Attention}(L^q(Q_T), L^k(K_T), L^v(V_T))), \tag{8}$$

where $L$ represents a linear projection. The output Z is then refined via a multi-layer perceptron (MLP), followed by layer normalization with residual connections:

$$Q_M = \text{LN}(\text{MLP}(Z) + Z) + Q_T, K_M = K_T \oplus V_T. \tag{9}$$

Next, the temporally-aware feature $Q_M$ is fused with the current frame's spatial information and projected to form query pairs $(q_k, q_v)$. Simultaneously, the combined memory $K_M$ is projected to obtain key-value pairs $(m_k, m_v)$. These are fed into the mask affinity module, where cross-attention is computed, and the final spatiotemporal representation is obtained by:

$$\text{output} = q_v \oplus \text{Attention}(q_k, m_k, m_v). \tag{10}$$

### 3.4 LOSS FUNCTION

We apply multi-level supervision using a hybrid loss that combines weighted binary cross-entropy (BCE) Ji et al. (2022) and weighted intersection over union (IoU) Rahman & Wang (2016). The total loss is defined as:

$$\mathcal{L}_{all} = \sum_{i=1}^{4} \frac{1}{2^{i-1}} \left( \mathcal{L}_{bce}^w(P_t^i, G) + \mathcal{L}_{iou}^w(P_t^i, G) \right), \tag{11}$$

where $P_t^i$ is the prediction at the $i$-th decoder stage, and $G$ is the ground truth.

## 4 EXPERIMENTS

### 4.1 DATASETS AND IMPLEMENTATION DETAILS

**Datasets.** We evaluate our proposed HFSTI-Net on the two polyp datasets, SUN-SEG Ji et al. (2022) and CVC-612 Bernal et al. (2015). (1) The SUN-SEG dataset contains a total of 49,136 frames from 285 sequences, divided into three subsets: the training set with 19,544 frames from 112 sequences; the SUN-SEG-Easy test set with 17,070 frames from 119 sequences; and the SUN-SEG-Hard test set with 12,522 frames from 54 sequences. (2) The CVC-612 dataset consists of 612 frames from

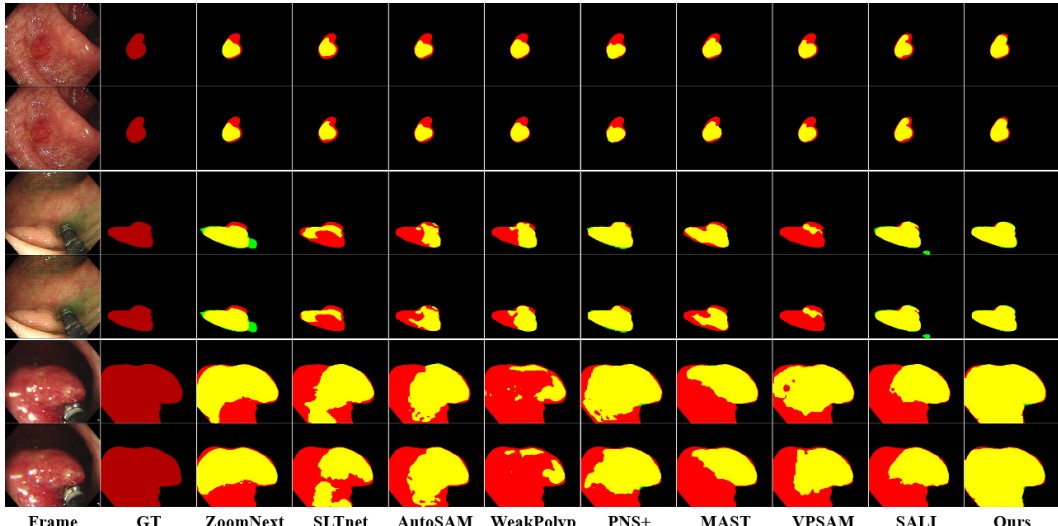

Figure 5: Visual comparison with SOTA methods on SUN-SEG. Red, green, and yellow represent ground truth, prediction, and their overlapping regions, respectively.

Table 4: Data visualization of module ablation on the SUN-SEG.

| HFSI | RMP | SUN-SEG-Easy | | | | SUN-SEG-Hard | | | |
|---|---|---|---|---|---|---|---|---|---|
| | | $S_a$ | $E_\varphi^{mn}$ | $F_\beta^\omega$ | Dice | $S_a$ | $E_\varphi^{mn}$ | $F_\beta^\omega$ | Dice |
| | | 89.51 | 92.93 | 83.72 | 86.20 | 88.03 | 92.29 | 80.62 | 83.67 |
| | ✓ | 89.71 | 93.68 | 84.15 | 87.04 | 88.59 | 92.91 | 81.24 | 84.03 |
| ✓ | | 90.51 | 93.87 | 84.53 | 87.24 | 88.97 | 93.02 | 81.86 | 85.27 |
| ✓ | ✓ | **90.73** | **94.86** | **85.82** | **88.03** | **89.63** | **93.92** | **83.26** | **86.27** |

Table 5: Sub-component ablation of HFSI on SUN-SEG.

| FFB | SRB | IFB | SUN-SEG-Easy | | | | SUN-SEG-Hard | | | |
|---|---|---|---|---|---|---|---|---|---|---|
| | | | $S_a$ | $E_\varphi^{mn}$ | $F_\beta^\omega$ | Dice | $S_a$ | $E_\varphi^{mn}$ | $F_\beta^\omega$ | Dice |
| | | | 89.71 | 93.68 | 84.15 | 87.04 | 88.59 | 92.91 | 81.24 | 84.03 |
| ✓ | | | 89.93 | 93.76 | 84.18 | 87.27 | 89.37 | 93.26 | 83.16 | 85.90 |
| | ✓ | | 90.09 | 94.07 | 84.43 | 87.28 | 88.92 | 93.18 | 82.14 | 85.27 |
| ✓ | ✓ | | 90.52 | 94.34 | 84.75 | 87.53 | 89.08 | 93.31 | 82.36 | 85.53 |
| ✓ | ✓ | ✓ | **90.73** | **94.86** | **85.82** | **88.03** | **89.63** | **93.92** | **83.26** | **86.27** |

31 colonoscopy sequences. For SUN-SEG, we set aside 20% of the training set as the validation set. For CVC-612, the dataset was split into training, validation, and test sets with a 6:2:2 ratio.

**Evaluation metrics.** For a thorough comparison, we utilize four evaluation metrics: Dice, structure-measure ($S_\alpha$) Fan et al. (2017), enhanced-alignment measure ($E_\phi^{mn}$) Fan et al. (2021), and weighted F-measure ($F_\beta^w$) Margolin et al. (2014), following previous studies Ji et al. (2022); Hu et al. (2024).

**Implementation details.** The proposed network was implemented on the PyTorch Paszke et al. (2019) platform and trained on one NVIDIA 3090 GPU. $\mathrm{Pvtv2\_b2}$ Wang et al. (2022) is used as the backbone for all experiments. We trained our model for 30 epochs with a batch size of 8. A video clip of 2 frames with a patch size of $352 \times 352$ was fed into the network. We adopted the Adam optimizer with an initial learning rate of $1e^{-4}$ and a weight decay of 0.1 for 10 epochs.

## 4.2 COMPARISON WITH EXISTING METHODS

To demonstrate the superiority of our proposed method, we compare it with popular image- and video-level object/polyp segmentation methods on SUN-SEG-Easy, SUN-SEG-Hard, and CVC-612. The compared methods include ZoomNext Pang et al. (2024), SLTNet Cheng et al. (2022), AutoSAM Shaharabany et al. (2023), WeaklyPolyp Wei et al. (2023), PNS+ Ji et al. (2022), MAST Chen et al. (2024), VPSAM Fang et al. (2024) and SALI Hu et al. (2024). These methods can be categorized into four groups: (1) natural video segmentation (NVS), (2) IPS, and (3) VPS. All training parameters are controlled and set identically for consistency.

**Comparison with SOTA methods.** The comparison results between our method and above state-of-the-art methods on the SUN-SEG and CVC-612 are shown in Table 1. Under identical experimental settings, our method outperforms all state-of-the-art approaches across every metric.

**Visual comparison with SOTA.** We present a visual comparison of our approach with state-of-the-art methods in Figure 5. The figure showcases segmentation results on three challenging cases: (1) similar foreground and background (lower case), (2) a sequence of consecutive low-quality frames

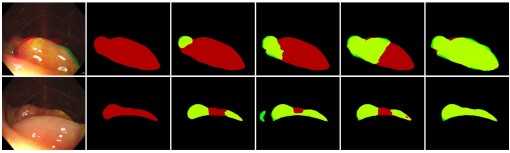
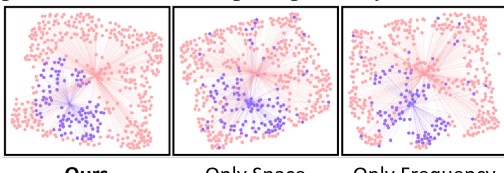

Frame  GT  w/o both  w/ HFSI  w/ RMP  w/ both

Figure 6: Visualization of module ablation on SUN-SEG. Red, green, and yellow denote GT, prediction, and overlap, respectively.

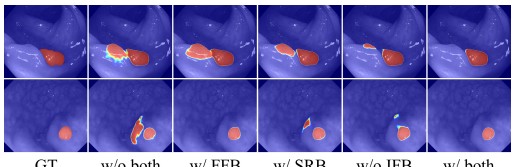

GT  w/o both  w/ FFB  w/ SRB  w/o IFB  w/ both

Figure 7: Visual comparison of image embedding on HFSI components in the SUN-SEG dataset.

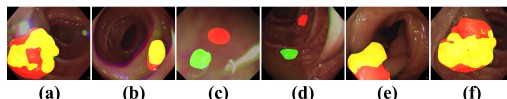

**Ours**    Only Space    Only Frequency

Figure 8: T-SNE visualization of different modeling strategies, where purple represents polyps and pink represents background.

(a) (b) (c) (d) (e) (f)

Figure 9: Failure cases. Red, green and yellow represent the GT, prediction and their overlapping regions, respectively.

(middle case), and (3) variations between consecutive frames (upper case). As shown in Figure 5, our method consistently achieves precise, stable segmentation where others struggle, demonstrating its effectiveness in challenging conditions.

**Performance efficiency comparison with SOTA.** The focus of this paper is to improve accuracy. To comprehensively analyze the strengths of our method, we also examine the trade-off between accuracy and efficiency. Our method achieves a real-time inference speed, with an FPS of 31.27, while surpassing the accuracy of other SOTA methods, as shown in Table 2.

Table 6: Ablation study on the SUN-SEG-Hard Dataset.

| Interaction Method | Dice (%) | Params (M) |
|---|---|---|
| Linear ($1 \times 1$ Conv) | 84.15 ($\downarrow$ 2.12) | 28.42 |
| Spatial Attention | 85.38 ($\downarrow$ 0.89) | 29.80 |
| **Ours** | **86.27** | **28.53** |

### 4.3 ABLATION STUDIES

**Effectiveness of RMP.** To assess the contribution of the recurrent mask-guided propagation (RMP) module, we performed ablation studies. Removing RMP leads to a clear drop in performance Table 4, as temporal cues are essential for consistent segmentation in video sequences. Visual results in Figure 6 show that RMP mitigates polyp fragmentation and improves boundary continuity across frames. We further evaluated the impact of varying the number of memory frames during training (1–4), as shown in Table 3. While using more frames improves accuracy, it slightly reduces FPS. Notably, even a single memory frame provides competitive performance with real-time inference, offering a practical trade-off between accuracy and efficiency.

**Effectiveness of HFSI.** As shown in Table 4 and Figure 6, HFSI improves both accuracy and boundary quality. To assess its components, we ablated FFB, SRB, and IFB individually. Results in Figure 7 and Table 5 show that removing IFB causes a notable drop, confirming the value of its interwoven fusion. Unlike simple merging, IFB enables bidirectional interaction between spatial and frequency domains for better global-local alignment. Further degradation occurs when disabling both IFB and SRB, and the worst performance appears when both FFB and IFB are removed, due to the lack of spectral filtering and fusion. These results validate the tightly integrated design of HFSI. As shown in Figure 8, shows that combining both frequency and spatial domains (Ours) leads to better separation of polyp and background classes compared to using only spatial or frequency information individually.

**Effectiveness of FFT/IFFT.** To verify the necessity of FFT for frequency–space interaction, we perform an ablation comparing FFT with a linear $1 \times 1$ convolution block and a spatial-attention

block (see Table 6). FFT naturally provides global context and isolates high-frequency components important for preventing "shape collapse," whereas linear and attention-based operators remain limited to spatial-domain processing. Results on SUN-SEG-Hard show that replacing FFT with a linear block leads to a clear performance drop ($-2.12\%$ Dice), and spatial attention also underperforms while using more parameters. These findings confirm that spectral interaction offers complementary structural cues that spatial operators cannot replicate.

**Impact of memory size N.** To determine the optimal number of historical frames stored in memory, we conduct an ablation study on $N$. As shown in Figure 11, the performance peaks when using $N = 8$, where the model effectively leverages temporal context without accumulating excessive noise. Increasing $N$ further (e.g., to 12) leads to accuracy degradation (Dice 84.31%) due to error accumulation. Thus, $N = 8$ achieves the best trade-off between temporal richness and memory reliability.

**Impact of input clip.** As shown in Figure 10 (a-b), we also explore the impact of different clip length $L$. The performance improves greatly when L increases from 1 to 2 because more spatio-temporal information is obtained. However, larger L values (e.g., 5 and 7) cause performance degradation. Longer clips can theoretically bring more spatio-temporal information, which is effective for clips composed of frames with high boundary discrimination. However, for colonoscopic videos with low boundary discrimination, we analyze the possible reason is that establishing spatio-temporal information between frames with a long temporal distance may bring redundant information that interferes with effective spatio-temporal information.

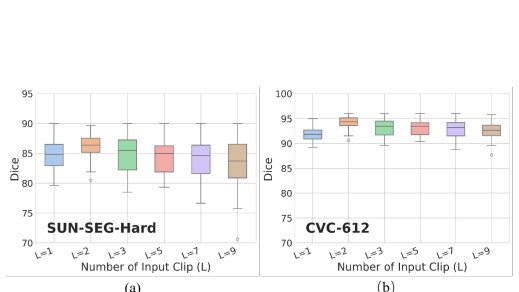

Figure 10: Ablation study on input clip.

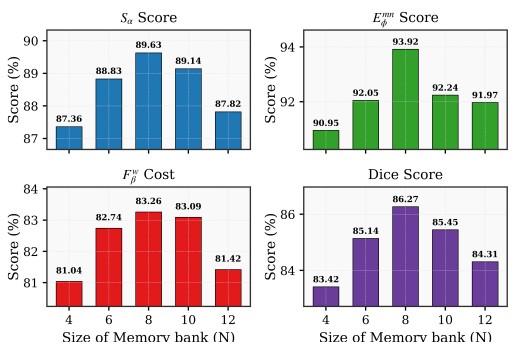

Figure 11: Ablation study on memory size.

## 4.4 DISCUSSIONS AND LIMITATIONS

While we only conduct experiments on colonoscopy video datasets, we believe that our HFSTI-Net is general enough to analyze other medical videos with similar challenges. Moreover, our methodology still exhibits certain limitations that warrant further investigation. As shown in Figure 9, spot interference (a-b), small polyps with very low contrast (c-d), and dramatic shapes (e-f) may limit our method.

## 5 CONCLUSION

In this paper, we propose the (*HFSTI-Net*), a novel network to tackle the critical challenges of shape collapse and episodic amnesia in video polyp segmentation. Its architecture effectively integrates three domains: the Hierarchical Frequency-spatial Interaction (HFSI) module leverages interwoven fusion of frequency and spatial cues to ensure fine-grained boundary localization and prevent shape collapse, while the Recurrent Mask-guided Propagation (RMP) module forms a stable temporal memory to maintain long-term consistency. These pathways are synergistically fused, yielding segmentations that are both precise and temporally coherent. Paired with efficient inference capabilities, our method achieves a superior balance between performance and real-time application. Extensive experimental results on SUN-SEG and CVC-612 demonstrate the effectiveness of our proposed method.

ACKNOWLEDGMENTS

This work was supported partly by National Natural Science Foundation of China (No. 62273241), Natural Science Foundation of Guangdong Province, China (No. 2024A1515011946), the Shenzhen Research Foundation for Basic Research, China (No. JCYJ20250604181940054), and a grant under Hong Kong RGC Collaborative Research Fund (project no C5055-24G).

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

## A  APPENDIX

### A.1  MORE ABLATION STUDIES ON SUN-SEG

**More comparisons on efficiency.** To provide a more comprehensive comparison of efficiency and accuracy, as shown in Figure 12, we further evaluate our method against other approaches. The results demonstrate that our method not only achieves real-time inference but also reaches state-of-the-art accuracy.

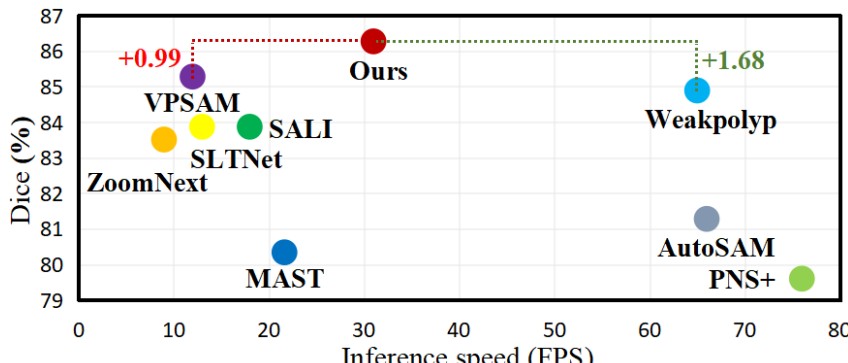

Figure 12: Performance-effciency comparison with other state-of-the-art methods on SUN-SEG.

**Component ablation of CVC-612.** We validated the effectiveness of our module design through ablation experiments on the CVC-612 dataset. As shown in Table 7, the results demonstrate that each module contributes individually, and the integrated framework outperforms existing methods, confirming the component effectiveness within our architecture. This validates the modules' generalizability across heterogeneous datasets.

**Performance-effciency of subcomponent ablation.** In addition to comparing the efficiency and performance with other SOTA methods, as shown in Table 8, we conducted a module-wise ablation study to analyze the trade-off between segmentation accuracy (Dice) and processing speed (FPS). The table's indicators further confirm that in automatic video polyp segmentation, HFSI and RMP address different challenges: HFSI decouples frequency-spatial features, while RMP ensures spatiotemporal consistency across frames. Their synergy enhances segmentation accuracy but impacts real-time performance. In summary, HFSI enhances feature representation, while RMP stabilizes temporal consistency, balancing global perception and local adjustment. Although this multidimensional interaction reduces processing speed, the decrease in efficiency is acceptable given the improved performance.

Table 7: Component ablation experiments on the CVC-612 Dataset.

| HFSI | RMP | CVC-612 | | | |
|------|-----|---------|---------|-------------|------|
| | | $S_a$ | $E_\varphi^{mn}$ | $F_\beta^\omega$ | Dice |
| | | 93.87 | 96.73 | 90.92 | 92.43 |
| | ✓ | 94.01 | 97.55 | 91.14 | 93.63 |
| ✓ | | 94.35 | 97.95 | 92.64 | 93.74 |
| ✓ | ✓ | **95.02** | **98.86** | **93.58** | **94.31** |

Table 8: Subcomponent ablation performance-efficiency comparison on SUN-SEG-Easy Hard set with $352 \times 352$ resolution.

| Method | SUN-SEG-Hard | | | |
|--------|------|--------|-----------|------|
| | Dice | GFlops | Param.(M) | FPS |
| w/o both | 83.67 | 31.96 | 28.16 | 37.39 |
| w/o HFSI | 84.03 | 45.86 | 28.23 | 35.27 |
| w/o RMP | 85.27 | 32.87 | 28.47 | 32.73 |
| w/ both | 86.27 | 46.77 | 28.53 | 31.27 |

### A.2  MORE VISUAL COMPARISON RESULTS

To demonstrate the superiority of our proposed method, we conduct visual comparisons with eight state-of-the-art methods on CVC-612, including FSEL Sun et al. (2024), MSCAF Liu et al. (2023), ZoomNext Pang et al. (2024), SLTNet Cheng et al. (2022), AutoSAM Shaharabany et al. (2023), WeaklyPolyp Wei et al. (2023), PNS+ Ji et al. (2022), and SALI Hu et al. (2024). As shown in Figure 13, the visual results demonstrate that our method outperforms previous state-of-the-art ap-

proaches in polyp boundary segmentation, integrity preservation, and localization accuracy. This demonstrates the method's effectiveness on both SUN-SEG and CVC-612 datasets.

To illustrate the segmentation capability on continuous video streams, we sample four consecutive frames at varying scales from the SUN-SEG dataset Ji et al. (2022) and compare our results with several state-of-the-art methods, as shown in Figure 14.The SUN-SEG comparison clearly shows that our method delivers more stable and consistently accurate segmentation across consecutive frames than existing state-of-the-art approaches. Despite the dataset's wide variations in lighting, texture, and object shapes, the red, green, and yellow regions in the figures show our predictions closely match the ground truth, demonstrating our approach's robustness.

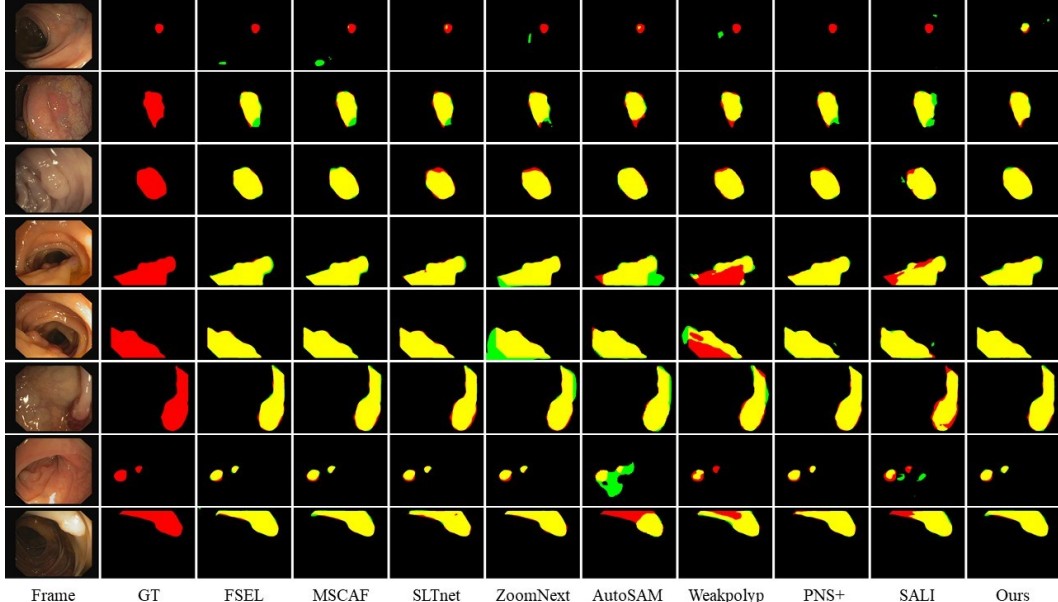

Figure 13: Visualization of module ablation on CVC-612 test set. Red, green and yellow represent the GT, prediction and their overlapping regions, respectively.

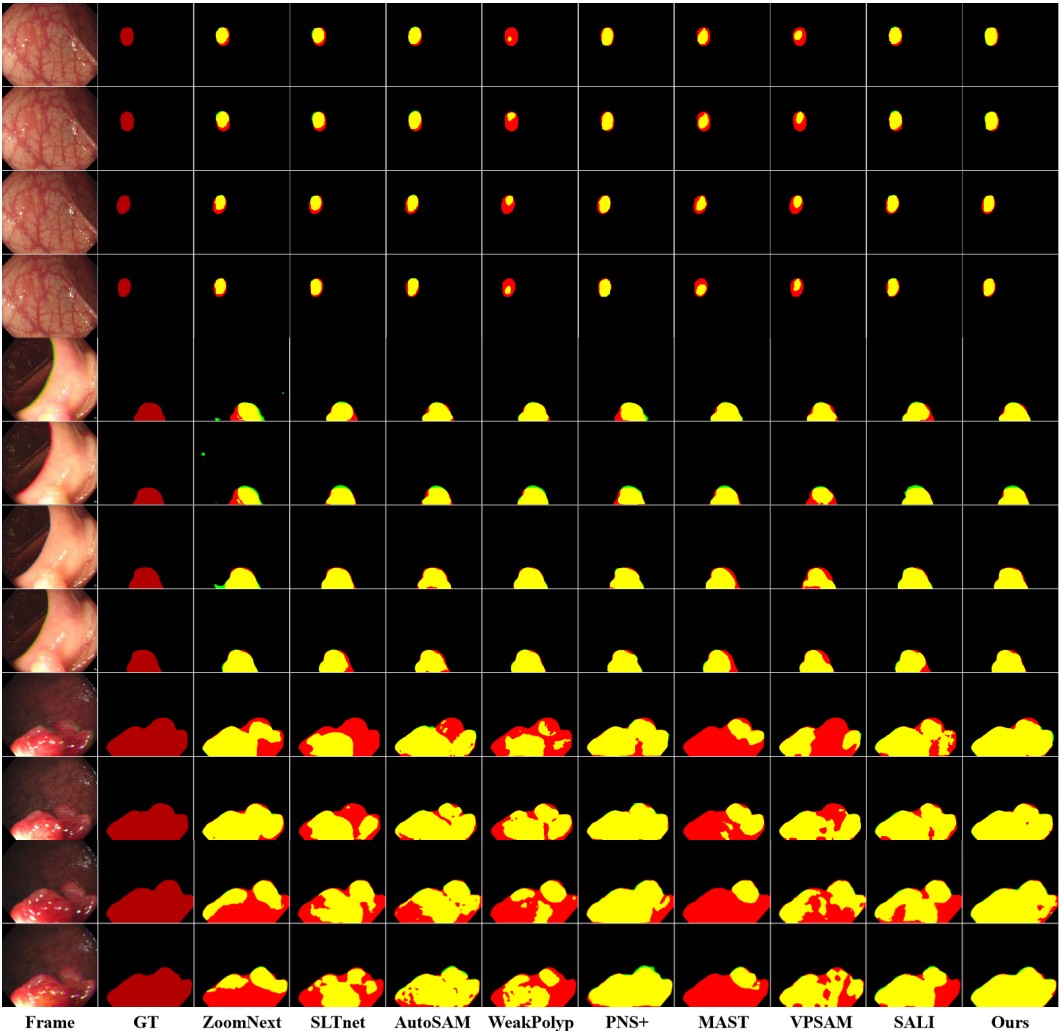

Figure 14: Visual comparison with SOTA methods on SUN-SEG. Red, green, and yellow represent ground truth, prediction, and their overlapping regions, respectively.

