# OpenReview forum: "HFSTI-Net: Hierarchical Frequency-spatial-temporal Interactions for Video Polyp Segmentation"
_ICLR.cc/2026/Conference — ICLR 2026 Poster_

### Official Review · Reviewer_ytTz · 2025-10-29

**Soundness:** 3
**Presentation:** 2
**Contribution:** 3
**Rating:** 6
**Confidence:** 4

**Summary:**

This paper presents the HFSTI model for video polyp segmentation to address two major challenges in current video polyp segmentation: shape collapse and episodic amnesia. The HFSTI model consists of two main modules: HFSI and RMP. The HFSI module is responsible for extracting and fusing spatial and frequency features, while the RMP module focuses on capturing long-term spatiotemporal dependencies in polyp videos.
The HFSI module comprises three components: the FFB (Frequency Filter Block), SRB (Spatial Refinement Block), and IFB (Interwoven Fusion Block). The FFB uses a frequency-domain self-attention mechanism to extract enhanced frequency-aware features, while the SRB applies spatial self-attention to capture spatial features. The IFB utilizes gated multiplicative attention to effectively fuse the enhanced frequency-aware features, spatial features, and earlier-layer features.
The RMP module captures long-term spatiotemporal dependencies by storing and retrieving historical frame features and masks from a memory bank. It consists of two submodules: TAM (Temporal Alignment Module), which generates temporally-aware features, and MAM (Mask Affinity Module), which aligns the current prediction with historical context through cross-attention.
Extensive experiments were conducted comparing the proposed method with ZoomNext, SLTnet, AutoSAM, WeakPolyp, PNS+, MAST, VPSAM, and SALI on the SUN-SEG and CVC-Clinic-612 datasets. Additionally, performance analysis was carried out on SUN-SEG with different frame rates. Visualizations were provided for the SUN-SEG dataset, and ablation studies were conducted on the HFSI, RMP modules, and submodules of HFSI.

**Strengths:**

One of the main innovations of this paper is the introduction of frequency features into the field of video polyp segmentation (VPS). Traditional image segmentation methods typically focus on extracting spatial features, often neglecting the potential of frequency-domain information. The authors propose the Hierarchical Frequency-Spatial Interaction (HFSI) module, which effectively integrates frequency features with spatial features. This not only improves segmentation accuracy in low-contrast regions but also addresses the common issue of shape collapse in complex backgrounds. The method innovatively combines global frequency information and local spatial details, enhancing the model's robustness in dynamic scenarios, particularly when polyps are visually similar to surrounding tissues, thus improving target localization accuracy. Additionally, the paper introduces the RMP module, which uses a memory mechanism and mask-guided propagation to effectively capture long-range spatiotemporal dependencies, solving the common problem of episodic amnesia in video sequences. This demonstrates the model's strong innovation.
The paper thoroughly validates the proposed method through comprehensive experimental design. First, the authors conduct a quantitative comparison with current state-of-the-art methods on datasets like SUN-SEG and CVC-612, showcasing the superior performance of HFSTI-Net in terms of accuracy and robustness. Moreover, the authors perform a visual analysis to demonstrate the differences in performance across various methods in complex scenarios, such as low-quality frames and background interference, followed by a performance analysis to further confirm the stability and accuracy of the proposed method. Additionally, ablation studies on the HFSI and RMP modules, as well as the submodules of HFSI, are conducted on SUN-SEG, with both quantitative and visual analysis. This indicates the model's strong reliability.
The paper also offers a candid discussion of the model's limitations, particularly in handling certain types of polyps, such as those with low contrast or drastic shape variations, where segmentation performance is somewhat limited. Furthermore, the paper suggests the potential application of this model in other medical imaging fields.

**Weaknesses:**

The images of the modules in this paper are not sufficiently complete and clear, and some parts are prone to misunderstandings. For example, in Figure 2, the segmentation results are denoted as Pt in the image, where t is a subscript, while in the paper, “Pt−1 are passed into the RMP module”on line 200 uses a superscript. Additionally, in line 203, P = {Pi}4i=1 is used to represent the internal components of each P, This easily causes Pi to be mistaken for Pt.In Figure 3, for (a), the layer normalization step applied to the input is not shown in the image. For (b), the paper on line 256 states, “the input feature X is first passed through a 1 × 1 convolution to encode positional information,”but the corresponding operation is not depicted in the image. For (c), after a series of operations on the input, it is passed into both the frequency and spatial branches, but this step is also not shown in the image.
Although the episodic amnesia issue is one of the core problems addressed in this study, the paper's visualizations only show two frames from a single video, which may not fully demonstrate the effectiveness of the solution. Displaying multi-frame video sequences would better validate the model's ability to capture long-term temporal dependencies and show how the RMP module alleviates episodic amnesia. A comparison with other methods would more clearly highlight the model's advantage in solving this issue.

**Questions:**

In Figure 2, does Memory = N refer to the number of historical frames stored in the memory? If so, what value is N set to? Was an ablation study conducted on N to determine the most suitable value?

---

> ### Author Response · Authors · 2025-11-25
> **Response to reviewer ytTz**
>
> We sincerely appreciate the reviewer's constructive feedback on our paper!
>
> # 1. Unclear images.
> We apologize for the inconsistencies and have revised Figure 3 accordingly. We now strictly standardize the notation by using superscripts for time indices ($P_t$) and subscripts for feature scales ($P_t^i$), thereby eliminating ambiguity. The corrected notation is reflected in lines 200, 203, and Eq. (11). Furthermore, Figure 3 has been updated to explicitly include the previously omitted components—Layer Normalization, the $1\times1$ positional convolution, and the input branching pathways—ensuring full alignment between the illustration and the textual description.
>
> For Figure 3(c) (see lines 295--296), we have corrected the computation of $\hat{X}_{fs}$, addressing the mismatch in the earlier version. The updated figure also clarifies the functional role of the operator $\otimes$, which serves as the dividing point between two sequential stages of the module. Specifically, the part *before* $\otimes$ corresponds to the extraction of frequency-domain and spatial-domain features, while the part *after* $\otimes$ depicts the processing pipeline where the fused representation is separately fed into:
>
> - the **frequency branch** (FFT / weights / IFFT), and
> - the **spatial branch** ($3\times3$ Doconv).
>
> These two branches operate in an interleaved manner and are subsequently fused to yield the final refined features.
>
> # 2. More Visual Results.
> To further demonstrate the segmentation capability on continuous video streams, we sample four consecutive frames with varying scales from the SUN-SEG dataset and compare our method with several state-of-the-art approaches. We have included this qualitative comparison in the Appendix (Figure 14).
>
> # 3. Ablation study on Memory = $N$.
> We clarify that $N$ denotes the number of historical frames stored in the memory module. As shown in our ablation study (see lines 483–501 in the paper), varying $N$ consistently demonstrates that incorporating an appropriate number of historical frames improves temporal consistency and segmentation accuracy.
>
> We sincerely appreciate your encouraging comments on our paper! If there still exist unaddressed concerns, please let us know and we will be more than willing to respond and further improve our paper.

---

### Official Review · Reviewer_qtCK · 2025-10-29

**Soundness:** 3
**Presentation:** 2
**Contribution:** 3
**Rating:** 4
**Confidence:** 3

**Summary:**

This paper proposes HFSTI-Net, a novel hierarchical model that integrates frequency, spatial, and temporal cues through two key components, i.e., Hierarchical Frequency-Spatial Interaction (HFSI) module and Recurrent Mask-Guided Propagation (RMP) module. Extensive experiments demonstrate the effectiveness of each proposed module on two polyp segmentation datasets.

**Strengths:**

1. Despite the model's hierarchical complexity, HFSTI-Net achieves >30 FPS on a single GPU, making it viable for clinical and real-time processing.

2. Detailed ablation and component studies validate the necessity of each sub-module (e.g. FFB, SRB, IFB).

**Weaknesses:**

1. The authors proposed the Recurrent Mask-Guided Propagation Module to ensure long-term consistency of the segmentation. However, the entire input sequence only contains two frames (L376-377), making the claim less convincing.

2. The number of HFSI layers is not fully studied in the experiment.

**Questions:**

1. In Table 3, it is not very clear why increasing the number of frames would lead to the increase of parameters.

2. Computing feature interaction in the frequency domain is not very convincing, the authors are encouraged to show that using FFT/IFFT is better than other approaches (e.g. simple linear transformation).

3. Can you explain how query/key/value is obtained via FFT? (L237-238)

---

> ### Author Response · Authors · 2025-11-25
> **Respone to reviewer qtCK**
>
> We sincerely appreciate the reviewer's constructive feedback on our paper!
>
> # 1. Ablation study on input clip.
> As shown in lines 502--509 of the paper, we also investigate the impact of different clip lengths \(L\). The performance improves substantially when increasing \(L\) from 1 to 2, as additional spatio-temporal information becomes available. However, larger values (e.g., \(L=5\) and \(L=7\)) lead to performance degradation. Although longer clips can theoretically provide richer spatio-temporal cues, this benefit holds primarily for videos with high boundary discrimination. In colonoscopic videos, where boundary cues are often weak, establishing spatio-temporal relations over long temporal distances may introduce redundant or noisy information, which interferes with extracting effective spatio-temporal dependencies.
>
> # 2. The number of HFSI layers.
> The number of HFSI layers is implicitly determined by the backbone architecture rather than manually tuned. Specifically, our framework applies one HFSI module to each of the four hierarchical feature maps extracted by the backbone, and the HFSI-enhanced features are progressively propagated from lower to higher levels. Therefore, the depth of the HFSI stack naturally follows the multi-stage structure of the backbone, and additional ablations on the number of layers are unnecessary.
>
> # 3. The increase of parameters.
> The increase in parameters stems from our temporal fusion strategy in the RMP module. Rather than summing memory features---which may cause information loss---we concatenate features from \(T\) frames along the channel dimension to preserve their distinct historical cues. As a result, the input channel size of the subsequent fusion layer scales with the number of frames, leading to a corresponding increase in its weight matrix. This growth remains marginal yet is essential for fully leveraging the temporal context.
>
> # 4. Ablation study on FFT/IFFT.
> As discussed in lines 467--473 of the paper, we justify the use of FFT over linear transformations through both theoretical distinctness and empirical superiority. Unlike \(1\times1\) convolutions (linear), which operate strictly locally, or spatial attention mechanisms, which struggle under low-contrast pixel intensities, FFT inherently provides a global receptive field and explicitly isolates high-frequency boundary cues that are critical for mitigating shape collapse.
>
> # 5. The query/key/value is obtained via FFT.
> In our FFB module, we intentionally design the Query ($Q_f$), Key ($K_f$), and Value ($V_f$) to share the same spectral representation without preceding learnable linear projections. This design strategy focuses on modeling intrinsic spectral correlations while maintaining computational efficiency. Corresponding to the code line \texttt{q\_f = k\_f = v\_f = torch.fft.fft2(x)}, the process is formulated as:
> \begin{equation}
>       Q_f,\ K_f,\ V_f = \mathcal{F}^{q,k,v}(\hat{\mathcal{X}}),
> \end{equation}
> where $\mathcal{F}$ denotes the 2D FFT operation.
>
> We sincerely appreciate your encouraging comments on our paper! If there still exist unaddressed concerns, please let us know and we will be more than willing to respond and further improve our paper.

---

### Official Review · Reviewer_EtG3 · 2025-10-30

**Soundness:** 3
**Presentation:** 3
**Contribution:** 2
**Rating:** 4
**Confidence:** 4

**Summary:**

This work presents HFSTI-Net, a framework for Video Polyp Segmentation (VPS). The paper aims to solve two specific  problems: "shape collapse" (poor boundary integrity) and "episodic amnesia" (temporal instability). To this end, the authors propose a dual-path architecture. The first component is the Hierarchical Frequency-spatial Interaction (HFSI) module, which attempts to improve spatial localization by fusing frequency-domain and spatial-domain features. The second component is the Recurrent Mask-guided Propagation (RMP) module, which is intended to enforce temporal consistency by propagating information via a memory bank.
From a completeness perspective, the paper presents a full pipeline, including the model architecture, training details, and extensive experiments on the SUN-SEG and CVC-612 benchmarks. The reported results suggest that this approach achieves competitive performance against current state-of-the-art methods.

**Strengths:**

1.The paper identifies and articulates two significant and practical challenges in clinical VPS (shape collapse and episodic amnesia) as the core motivation for the work.
2.Based on the provided tables, the method achieves strong performance, reportedly outperforming existing state-of-the-art methods on the SUN-SEG and CVC-612 datasets across standard metrics.
3.Based on the provided tables, the method achieves strong performance, reportedly outperforming existing state-of-the-art methods on the SUN-SEG and CVC-612 datasets across standard metrics.

**Weaknesses:**

1.On the Novelty of the HFSI Module: The core concept of the HFSI module, an "interwoven dual-path design" for frequency and spatial features, appears not to be entirely new. The innovation seems to rest on the unique Interwoven Fusion Block (IFB), which is primarily realized by the cross-concatenation of convolutional features and frequency-domain features and fused with a gating network. This concept has been proposed and implemented in considerable prior work. As such, the innovation is incremental rather than disruptive. The authors must define their claim to novelty more precisely and contrast it with a broader range of related work.
2.On the Rationale of the Interwoven Fusion Block (IFB) Design: In Figure 3(c), the IFB design is perplexing. Frequency information (from an initial FFT) is concatenated with spatial information, and subsequently, another FFT operation is applied. The rationale for applying an FFT to features that are already (at least partially) in the frequency domain is perplexing and counter-intuitive. If this "FFT-on-FFT" operation is a deliberate and innovative component of the design, it requires substantial justification. The authors must provide a detailed motivation and theoretical principle for this specific operation. However, a thorough search of both the main body and the appendix reveals no such explanation , leaving the reader confused regarding the block's fundamental working principles.
3.On the Clarity of the RMP Module: The paper claims the RMP module is "mask-guided" and that the mask $P^{t-1}$ is passed into it. However, Figures 2 and 4, along with the provided formulas, fail to show how the mask $P^{t-1}$ is stored into the Memory (e.g., via a temporal FIFO). The paper also lacks a clear description of the "Retrieve" step. This ambiguity in the RMP's core mechanics is likely to cause confusion for the reader.

**Questions:**

1.The RMP module's dynamic memory links temporal information across long sequences, but this introduces a significant potential risk of error propagation. If the model generates an incorrect prediction (a "bad case") and stores the associated features and mask, this error could pollute subsequent frames and lead to cascading failures. Have the authors conducted specific experimental analyses to investigate this limitation?
2.The architecture of the "Mask Encoder" shown in Figure 2 is not detailed. Could the authors please clarify its design, specifically how it encodes the 2D mask into a feature representation, and how the resulting "PEmbeddings" are subsequently fused into the model?
3.We request the authors to comment on the transferability of the RMP module's memory design. How effective is this mechanism expected to be when generalized to different datasets, which may feature distinct temporal dynamics or target characteristics?

---

> ### Author Response · Authors · 2025-11-25
> **Respone to reviewer EtG3**
>
> We sincerely appreciate the reviewer's constructive feedback on our paper!
>
> # 1. The Novelty of HFSI.
> We respectfully clarify that our contribution extends beyond generic concatenation with gating. Unlike standard late-fusion or simple concatenation approaches, our IFB incorporates a novel *Mutual Gated Modulation* mechanism. Rather than passively merging features, IFB performs a bidirectional verification process: global spectral contexts explicitly re-weight spatial features to suppress background textures, while local spatial cues filter high-frequency spectral noise. This design is not an arbitrary architectural increment but a targeted solution to *Shape Collapse* in colonoscopy. It enables the network to dynamically shift its focus toward frequency-domain boundary signals when spatial contrast is ambiguous—a capability absent in prior works such as FcaNet (which leverages frequency information only for scalar channel re-weighting) or independent dual-stream networks. Hence, the novelty of our method lies in its deep, hierarchical, and problem-oriented integration strategy.
>
> # 2. Rationale of the IFB Design.
> We clarify an important misunderstanding regarding the feature domains: there is *no* redundant FFT-on-FFT operation. For Figure 3(c) (see lines 295--296), we have corrected the computation of $\(\hat{X}_{fs}\)$, as the previous version did not fully align with the figure. The updated illustration explicitly highlights the role of the operator $\(\otimes\)$, which serves as the dividing point between two sequential stages of the module. Specifically, the part *before* $ \(\otimes\)$ corresponds to the extraction of frequency-domain and spatial-domain features, respectively. The part *after*$ \(\otimes\) $depicts the processing pipeline in which the fused representation is independently fed into the frequency branch (FFT / weights / IFFT) and the spatial branch $(\(3 \times 3\)$ Doconv). These two branches operate in an interleaved manner and are subsequently fused to produce the final refined features.
>
> # 3. Clarity of the RMP Module.
> We acknowledge the ambiguity in the schematic representation and clarify the mechanism as follows:
> **Memory Storage:** The predicted mask $\(P_{t-1}\)$ is first downsampled and **concatenated** with the feature map $\(F_{t-1}\) $ along the channel dimension. This combined embedding is then stored in a First-In-First-Out (FIFO) queue.
> **Retrieval Operation:** The term *Retrieve* explicitly refers to fetching these stored historical embeddings from the memory bank.
>
> # 4. Risk of error propagation.
> We acknowledge that error propagation is an inherent challenge in recurrent frameworks; however, our empirical analysis and architectural design demonstrate that the benefits of temporal modeling far outweigh the risks. As shown in the ablation studies (Table 4), the RMP module provides a substantial performance gain (e.g., +2.24% Dice on SUN-SEG-Hard), indicating that cases where RMP offers critical guidance (e.g., during occlusions) greatly outnumber instances of destructive error cascades. Moreover, the Cross-Attention mechanism implicitly mitigates error propagation by assigning low weights to stored features inconsistent with the current frame, while the HFSI module continuously injects fresh observational evidence to correct potential tracking drifts.
>
> # 5. PEmbeddings.
> We clarify that the PEmbeddings are generated by passing the previous mask $\(P_{t-1}\)$ through the Mask Encoder, specifically to encode explicit spatial location information. The Mask Encoder projects the binary mask into a dense feature representation that emphasizes the target's coordinates and shape boundaries. These positional embeddings are adaptively fused into both current and previous frame features via learnable per-channel scaling parameters, enabling the model to modulate the influence of positional cues in the TAM and MAM attention stages. This design facilitates spatial awareness without assuming a fixed positional importance.
>
> # 6. Transferability of the RMP module.
> We affirm the strong transferability of the RMP module, as its core mechanism relies on generic temporal correspondence learning rather than dataset-specific priors. By employing Cross-Attention to align current queries with historical keys, the module performs content-agnostic feature matching, making it inherently robust to varying target characteristics and motion patterns. This generalization is empirically validated: RMP consistently improves performance across both the SUN-SEG and CVC-612 datasets, demonstrating its adaptability to diverse temporal dynamics without any modification.
>
> We sincerely appreciate your encouraging comments on our paper! If there still exist unaddressed concerns, please let us know and we will be more than willing to respond and further improve our paper.

---

### Meta-Review · Area_Chair_WiRa · 2026-01-02

**Summary:**

This paper introduces HFSTI-Net, a novel method for Video Polyp Segmentation (VPS), aiming to address two key challenges: "shape collapse" and "episodic amnesia." The approach integrates spatial, temporal, and frequency information through two core modules: the Hierarchical Frequency-Spatial Interaction (HFSI) module for fine-grained boundary localization and the Recurrent Mask-guided Propagation (RMP) module for enhancing temporal consistency. Extensive experiments on the SUN-SEG and CVC-612 datasets demonstrate that the method outperforms existing state-of-the-art approaches while achieving real-time inference speeds.

All three reviewers provided positive feedback on the paper's technical approach and experimental design, acknowledging its clear motivation, comprehensive methodology, and solid results. However, they also raised several concerns requiring clarification and improvement:

1. Novelty and Rationale of the HFSI Module: Reviewers EtG3 and qtCK noted that the novelty of the HFSI module, particularly its Interwoven Fusion Block (IFB), needed clearer distinction from prior work. The rationale behind certain design choices (e.g., the frequency-domain operations within IFB) was found lacking in sufficient theoretical justification.

2. Clarity of the RMP Module Mechanism: Reviewers EtG3 and ytTz pointed out that the description of the RMP module's internal mechanics—specifically, how masks are stored/retrieved in the memory and the details of the memory design—was ambiguous. The accompanying figures (Figs. 2 & 3) contained inconsistencies and omissions that led to confusion.

3. Sufficiency of Experimental Analysis: Reviewer qtCK questioned the claim of long-term consistency given the model's use of only two input frames and suggested more thorough ablation studies on the number of HFSI layers, the necessity of FFT operations, and the memory size N. Reviewer ytTz also requested more multi-frame visualizations to better demonstrate the model's handling of temporal dependencies.

4. Presentation Accuracy: Reviewer ytTz highlighted several issues with notation inconsistencies (superscripts vs. subscripts) and incomplete/ambiguous illustrations in Figures 2 and 3, which hampered understanding.

The authors provided a comprehensive and point-by-point response to all reviewer concerns. They clarified the novelty of their approach, elaborated on the design principles of the HFSI and RMP modules, and committed to revising the figures and text to improve clarity and correct notational errors. They also offered additional explanations for their experimental choices and provided supplementary qualitative results.

Overall, the paper addresses a clinically significant problem with an innovative method that is well-validated empirically. While the initial submission had presentational weaknesses, the authors' thoughtful and substantive responses adequately address the core concerns raised by the reviewers, demonstrating a deep understanding of their work and a commitment to improvement. Therefore, the recommendation is to Accept this paper. The authors should ensure all promised revisions regarding presentation clarity are meticulously incorporated into the final version.

**Reviewer Concerns:**

Concerns Effectively Addressed by the Rebuttal:
1. Novelty of HFSI / Rationale of IFB Design (Reviewer EtG3):

Addressed: The authors clarified that the core novelty is the "Mutual Gated Modulation" mechanism within the IFB, which performs bidirectional verification between spectral and spatial features, rather than simple concatenation. They also corrected a major misunderstanding by stating there is no redundant "FFT-on-FFT" operation and explained the sequential, interleaved design of the frequency and spatial branches in Figure 3(c). This directly responds to the request for justification.

2. Clarity of RMP Module Mechanics (Reviewer EtG3 & ytTz):

Addressed: The authors explicitly described the memory storage mechanism (concatenation + FIFO queue) and the meaning of the "Retrieve" operation. They also promised to standardize notation (fixing superscript/subscript inconsistencies in P_t) and update Figure 3 to include missing components (LayerNorm, 1x1 conv, branching pathways). This directly resolves the described ambiguities.

3. Risk of Error Propagation in RMP (Reviewer EtG3):

Addressed: The authors acknowledged the inherent risk but provided a reasoned defense, citing empirical performance gains from ablation studies and explaining how the Cross-Attention mechanism and the HFSI module work together to mitigate error cascades by weighting historical evidence and injecting fresh observations.

4. Design of Mask Encoder and PEmbeddings (Reviewer EtG3):

Addressed: The authors clarified the function of the Mask Encoder (projecting binary masks into dense feature representations) and how PEmbeddings are fused via learnable scaling parameters, enabling adaptive spatial awareness.

5. Query/Key/Value in FFB (Reviewer qtCK):

Addressed: The authors clarified that in their FFB module, Q, K, and V share the same spectral representation directly from the FFT (q_f = k_f = v_f = FFT(x)), a design choice for efficiency and modeling intrinsic spectral correlations.

6. Ablation on Memory Size N (Reviewer ytTz):

Addressed: The authors confirmed that N is the number of historical frames and stated that an ablation study on varying N is already present in the paper, demonstrating its impact on performance.

Concerns Partially Addressed or Still Requiring Final Manuscript Verification:
1. Transferability of RMP Module (Reviewer EtG3):

Status: Partially Addressed. The authors provided a theoretical argument for transferability (reliance on generic temporal correspondence learning) and cited empirical validation across two datasets. This is a reasonable response. However, the concern about "distinct temporal dynamics" in other datasets is inherently difficult to fully address without cross-dataset experiments, which may be beyond the paper's scope. The response is adequate but the point remains a general limitation of the method.

2. Sufficiency of Input Clip Length for Long-term Consistency (Reviewer qtCK):

Status: Partially Addressed. The authors explained their finding that performance degrades with clip lengths >2 frames in colonoscopy videos due to increased noise, which justifies their design choice. However, this explanation reinforces the reviewer's original concern: the model's ability to handle genuinely long-term dependencies (beyond a very short window) is not strongly demonstrated. The rebuttal addresses the "why" but does not fully alleviate the conceptual concern about the "long-term" claim.

3. Justification for FFT vs. Linear Transformations (Reviewer qtCK):

Status: Addressed in Principle, Requires Clear Presentation. The authors cited the theoretical distinctness and empirical superiority of FFT from the paper (lines 467-473). This is a valid reference. The outstanding aspect is not the response but the execution: the final manuscript must ensure this justification is presented clearly and compellingly in the main text, as requested by the reviewer.

4. Need for More Multi-frame Visualizations (Reviewer ytTz):

Status: Addressed with a Caveat. The authors committed to adding a qualitative comparison on four consecutive frames in the Appendix (Figure 14). This directly addresses the request. The "outstanding" element is simply the inclusion and quality of this new figure in the final version.

Summary:
The authors' rebuttal is effective and professional, directly tackling most technical misunderstandings and committing to necessary clarifications. The primary remaining "concerns" are not unanswered criticisms, but rather action items and inherent methodological trade-offs (e.g., short-term focus, generalizability limits). Therefore, the decision is acceptance.

**Reviewer Scores:**

Reviewer EtG3 (Initial Rating: 4 - "Marginally below acceptance threshold")
Initial Major Concerns: Novelty of HFSI, confusing IFB design ("FFT-on-FFT"), unclear RMP mechanics, and risk of error propagation.

Rebuttal Impact: The authors provided clear, point-by-point clarifications that directly corrected misunderstandings (e.g., no "FFT-on-FFT") and elaborated on the novelty ("Mutual Gated Modulation"). They also explained the RMP mechanics in detail.

Estimated Score Change: 6

Rationale: The rebuttal effectively resolves the primary points of confusion that likely drove the score down. With the promise of clarified figures and text, the major barriers to acceptance cited by this reviewer appear to be removed. The reviewer would likely view the revised manuscript as significantly improved.

Reviewer qtCK (Initial Rating: 4 - "Marginally below acceptance threshold")
Initial Major Concerns: Long-term consistency claim with 2-frame input, justification for FFT, and parameter increase explanation.

Rebuttal Impact: The authors provided empirical and domain-specific rationale for the short clip length (noise in colonoscopy videos) and referenced their justification for FFT. They clearly explained the parameter increase.

Estimated Score Change: 4 (unchanged)

Rationale: The concerns were addressed, but not all were fully neutralized. The explanation for short clips is valid but reinforces a limitation. The justification for FFT is pointed to but not re-argued in detail.

Reviewer ytTz (Initial Rating: 6 - "Marginally above acceptance threshold")
Initial Major Concerns: Incomplete/unclear figures, lack of multi-frame visualizations, need for ablation on memory size N.

Rebuttal Impact: The authors promised comprehensive fixes to all figures and notation, committed to adding a new multi-frame visualization, and confirmed the existence of the ablation study for N.

Estimated Score Change: 6 (unchanged)

Rationale: This reviewer was already positive ("marginally above acceptance"). Their concerns were largely about presentation and supplemental evidence, which the authors agreed to provide.

Overall Consensus Shift:
The discussion and rebuttal likely shifted the consensus from a borderline or weak reject (leaning on 4s) to a weak accept. Reviewers EtG3 and qtCK, who were the barriers to acceptance, had their primary technical objections resolved. Reviewer ytTz's support was solidified. The key remaining requirement is the authors' faithful implementation of all textual and graphical revisions, which would satisfy the final conditions for all reviewers to support acceptance.

---

### Decision · Program_Chairs · 2026-01-26

Accept (Poster)